# Empirical prediction of patent pledge financing of pharmaceutical enterprises—A case study in Jiangsu China

Xiaojuan Zhao[1], Yunhua Liu[1,2]*

1 School of Intellectual Property, Nanjing University of Science and Technology, Nanjing, Jiangsu, China,
2 University of Copenhagen, Copenhagen, Danmark

* liuyunhua2017@gmail.com

## Abstract

Financing by patent pledge is an important way for small- and medium-sized pharmaceutical enterprises to address financing problems. In this study, eight indexes are analyzed considering both the pledge patent value and pledger credit value. And a prediction model for the patent pledge financing amount for pharmaceutical enterprises is constructed for the first time using the analytic hierarchy process and the fuzzy comprehensive evaluation method. Three levels of financing amount are concluded through the prediction model and prediction results corresponding with the financing amount are displayed. This model was designed to help small- and medium-sized pharmaceutical enterprises get access to financing through patent pledge to relieve their financial stress. At the same time, it provides guides for pledgees and policymakers to improve the efficiency and quality of patent pledge. This work is reliable and valid in that it constructs this prediction model based on systematical data from official data sources.

## 1 Introduction

Recently technological innovation has been at the heart of the global economy, leading to the surge of patents. For technology-dependent enterprises, they could enhance the financing capability by making the best of patents, for example patent pledge. As the typical technology-dependent industry, pharmaceutical industry is of great significance to improve the quality of national life and the country's scientific and technological development. In recent years, there has been significant continuous development in pharmaceutical technology innovation with the support of the state. In 2017, there were 19,878 patent applications, including 10,886 invention patents. And there are 41,673 valid invention patents in China's pharmaceutical manufacturing in total [1]. Such considerable intangible assets should be transformed into economic strength to realize the benefits of innovation through implementation, licensing, transferring, and financing, effectively promoting the progress of the pharmaceutical industry and the development of the country [2].

Pharmaceutical enterprises have a large demand for capital and need a long industrialization period [3]. Capital plays a decisive role in the industrialization of technology and the

**Data Availability Statement:** All relevant data are within the manuscript and its Supporting Information files.

**Funding:** Yunhua Liu and this work is supported by Soft Science Foundation of the National Intellectual

Property Administration, PRC (No SS18-B-24), Intellectual Property Development & Research Center of Jiangsu and China Scholarships Council (No 201806845002). The funders had no role in study design, data collection and analysis, decision to publish, or preparation of the manuscript.

**Competing interests:** The authors have declared that no competing interests exist.

development of enterprises. With the development of market economy, there are an increasing number of small- and medium-sized pharmaceutical enterprises possessing extraordinary innovative potential and strong intellectual property advantages, which need a vast amount of money to conduct research and survive in the competing market. However, due to the lack of favorable commercial credit and traditional fixed assets for guarantee, it tends to be difficult for such enterprises to obtain financing. Thus, funding issues have become the main bottleneck restricting their development. Using intellectual property rights to turn intangible assets such as patents into funds can be an effective way to solve funding problems. In particular, patent pledge financing has the potential to accomplish this. However, patent pledge financing in China's pharmaceutical industry is still in the exploratory phase and a comprehensive system should be established.

Besides, with the high interest and assessment fees, the amount of pledged financing is often disproportionate to the patent value. Many pharmaceutical enterprises cannot control the amount of patent pledge financing. Sometimes they use high-value patents to obtain small loans, which is not helpful to solve the funding problem. Therefore, they need a system to evaluate the patent pledge financing amount preliminarily so that they can strive to match the financing amount and the pledge patent value.

Inspired by this gap, in this paper, we first analyze the situation of patent pledge and then constructed a patent pledge financing amount prediction model for pharmaceutical enterprises from the perspective of the pledge patent value and the credit value of the borrower using the patent pledge financing data of pharmaceutical enterprises in Jiangsu Province, China. The reason for choosing Jiangsu Province is that the biopharmaceutical industry in Jiangsu Province ranks first in China due to its regional characteristics and comprehensive innovation ability advantages. Eight indexes were analyzed and the analytic hierarchy process (AHP) and fuzzy comprehensive evaluation method were used. The prediction method was developed for patent pledge, aiming to help small- and medium-sized pharmaceutical enterprises obtain the ideal amount of financing through patent pledges to solve their financial problems. This work is reliable and valid in that it constructs this prediction model based on systematical data from official data sources for the first time. And the innovations are twofold. The first one is that we construct the model for predicting the patent pledge financing amount for the first time which can be used in practice. The other one is we use and integrated real and systematic data from official data sources.

## 2 Literature review

In previous literature, the main focus of patent pledge has been on defining, investigating the significance and evaluating the value of pledged patents in theory. Contreras summarized the range of recent development in the evolution of patent pledges, including the structural complexity and sophistication, the motivations and the trend toward democratization and internationalization of pledge behavior and so on [4]. Using records from the United States Patent and Trademark Office (USPTO), Mann has shown that companies which previously pledged patents exhibited a gradual increase in patenting output and the use of patents as collateral [5]. Chava et al. showed that borrowers with higher-quality patents receive lower spreads on bank loans [6]. Ehrnsperger et al. proposed a three-dimensional taxonomy that distinguishes eight types of patent pledge based on inductive research about 60 patent pledges [7]. Chen and Chang examined the relationship between four patent quality indexes and the market value of an enterprise in the pharmaceutical industry in the United States [8]. As for patent pledge financing in the biomedical industry, Ding et al. constructed an evaluation index system for patent pledge financing through seven indexes about financing object, financing content, and financing

subject, and also discussed the related policy issues [9]. Tang and Hong constructed a patent pledge value index system from three first-class indexes: the technical, economic, and legal value of a patent, which includes six technical value second-class indexes, such as technology maturity and industrialization feasibility; seven second-class indexes of economic value, such as market recognition and competitiveness; and eight legal value second-class indexes, such as stability and avoidance [10]. Most of the literature on patent pledges currently are only qualitative or theoretical analysis since patent pledge financing data could not be acquired from official public information. According to the literature, the number of patent pledge contracts has increased with the promotion of national and local policies. However, the amount of patent pledge financing is still low [11]. In addition, researchers showed that the evaluated value of pharmaceutical patents for pledge financing is far lower than their actual value because of the non-standard assessment approach and high uncertainty and risk of pharmaceutical industry [12].

## 3 Methodology

### 3.1 Data

The dataset used for the study was obtained from some different data sources in China, including Jiangsu Provincial Drug Administration, the official websites of enterprises, the national enterprise credit information publicity system, China National Intellectual Property Administration (CNIPA), and Jiangsu Provincial Intellectual Property Office. First of all, article 7 of the Drug Administration Law of the People's Republic of China stipulates that the establishment of a pharmaceutical enterprise should be approved by the drug regulatory authority of the people's government of the province, autonomous region, or municipality directly under the Central Government where the enterprise is located and issued to the Pharmaceutical Production License. Thus, from the first official dataset, Jiangsu Provincial Drug Administration, we obtained a list of pharmaceutical enterprises in Jiangsu Province. An enterprise's awards were obtained through its official website. The enterprise's credit was acquired from a range of reliable websites including the national enterprise credit information publicity system, the national unified social credit code information verification system, the local enterprise credit inquiry system, the People's Bank of China Credit Information Center, China Judgment online, the Intellectual Property Judgment online, and the China Executive Information online. The "Regulations on the Registration of Patent Rights Pledge" stipulates that patentees should register the pledged patents in CNIPA. Therefore, through the Patent Pledge Registration System of the CNIPA, we could determine if a Jiangsu pharmaceutical enterprise had pledged patents. Then, the characteristics of the pledged patent, such as the inventor of the pledged patent, the application time, and so on, could be obtained through the patent search system of CNIPA. The patent pledge financing amount was obtained by application to the Jiangsu Provincial Intellectual Property Office. The data was retrieved on March 28, 2019.

### 3.2 Method

First, the descriptive method was used to analyze the situation of patent pledge. Then according to previous research, indexes for predicting patent pledge financing amount are selected. As for evaluating the indexes, considering that the effects of the indexes on the patent pledge financing amount are different, and the evaluation of each indicator is mostly qualitative. Thus, it is difficult for an evaluator to provide specific and exact quantitative values. In addition, it is too subjective and arbitrary to evaluate by qualitative methods only by experience, as this results in a large margin of error. Based on this fact, in this study, the analytic hierarchy process (AHP) and fuzzy comprehensive evaluation method were used to evaluate the indexes and construct the model [13, 14]. Firstly, through empirical judgment, multiperson review,

and summary of references, the indexes were compared pairwise, which provided quantitative results of relative importance and formed a judgment matrix. Then, the weight of each index was calculated. After that, the fuzzy comprehensive evaluation method was used to evaluate the indexes. The basic idea of the fuzzy comprehensive evaluation method is to obtain a quantitative value for each index after standardization for the results of the fuzzy comprehensive evaluation. Then, the weight and score are combined and calculated to obtain the comprehensive evaluation result of the object [15]. To use the fuzzy comprehensive evaluation method, the evaluation levels, and corresponding standards should be determined first [16]. In this study, the levels and standards were set as K = {very good, good, medium, poor, very poor} = {9,7,5,3,1}. That means the subjective judgment could be expressed and processed in a quantitative form through the combination of qualitative analysis and quantitative calculation, which can solve the problem of ambiguity for patent value and has a relatively simple operation. With this method, the scientific validity, reliability, and feasibility of the prediction result of the patent pledge financing amount can be largely improved.

### 3.3 Index system for prediction model

In patent pledge financing, the value of a patent depends not only on the value of the pledged patent itself but also on the credit value of the pledger. Therefore, in this study, considering both the value of the pledge patent and the credit value of the pledger, we analyzed eight indexes to construct the prediction model for the patent pledge financing amount.

**3.3.1 Patent value.** One patent can be used for pledge financing, but in order to get a greater amount of financing, a patent portfolio is usually pledged as a whole. In this paper, the pledged patent refers to all patents in the contract. From the perspective of the patent value, five indexes were analyzed, including patent innovation, market application of patent, patent stability, the importance of patents in products, and scope of patent protection.

①Patent innovation: This refers to the contributions the patent makes to the related technology. Compared with existing technology, the more significant and advanced the patent, the better the innovation. Patent innovation is analyzed by interpreting the patent specifications and its references and citations. Generally, the more thoroughly a patent improves technology, the fewer the references and the greater the citations, indicating that the innovation is high and, correspondingly, the patent value is high.

②Market application of patent: This refers to the patent status. In other words, we measure it by investigating if the patent has been applied to the market or had market transformation ability. Specifically, it is evaluated by judgments of relative experts and analyzing the background of patent specifications, combined with searching and analyzing patents of related technologies. The better the patent market is, the more competitive it is, and the pledgee is more likely to realize maximum benefit protection if the borrower is unable to repay the loan.

③Patent stability: Stability mainly refers to the possibility of a patent being invalidated. It is reflected by the novelty, creativity, practicability, and claims of the patent. Further, it can be analyzed through the characteristics of the claim, the outcome of the process of invalidation, lawsuits and the transfer experience. Patent stability is a very important index in pledge financing. Once a patent is invalidated, the pledgee's rights cannot be guaranteed.

④The importance of patents in specific products: This is analyzed by whether the patent is a core patent and whether there is an alternative technical solution to solve the same or similar problems. If it is a core patent and there is no alternative technology, its pledge value is high. This index can be provided from the patent specification.

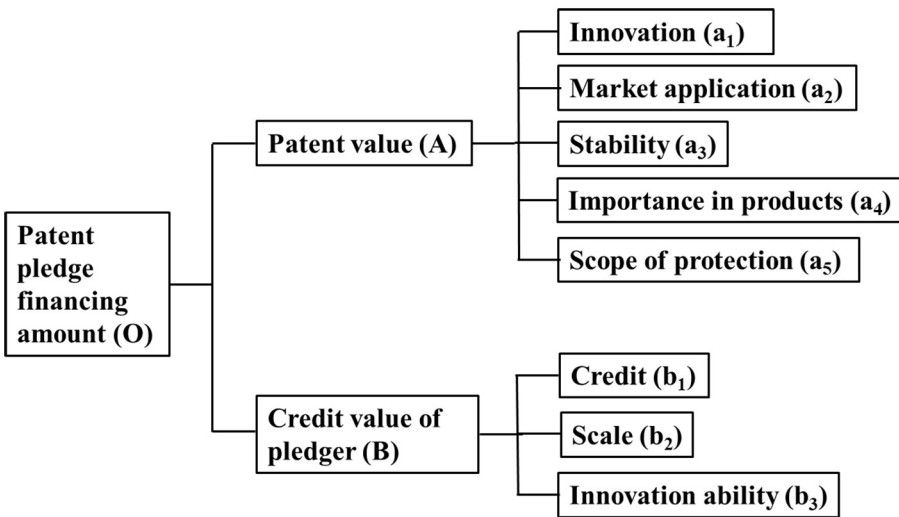

**Fig 1. Index system for predicting the patent pledge financing amount.**

⑤ The scope of patent protection: This refers to the technical scope, region, and time of patent protection. It is acquired by analyzing whether there is a patent family, the remaining time of patents, and the scope of protection of claims.

**3.3.2 Credit value of the pledger.** In pledge financing, the pledgee pays attention not only to the existing size of the pledger but also to its credit and development. For small- and medium-sized pharmaceutical enterprises, growth is determined by innovation. Therefore, in this paper, for pharmaceutical enterprises, we mainly considered the aspects of credit, scale, and innovation ability.

①Credit: Combining the patent pledge experience, the credit rating certificate from the enterprise's website and the credit publicity inquiry system at all levels, we analyze the enterprise's credit.

②Scale: We assessed the scale of an enterprise by establishment time, number of employees, registered capital, turnover, and so forth.

③ Innovation ability: The innovation ability was evaluated by the number of granted patents and the proportion of invention patents among all patents, combined with the obtained awards and certifications of intellectual property management systems. Science and technology awards were obtained from the official website of an enterprise.

Based on the above analysis, the index system for predicting the patent pledge financing amount is shown in Fig 1.

# 4 Results

## 4.1 Description analysis

There are 561 pharmaceutical enterprises in Jiangsu Province. However, the data of the Patent Pledge Registration System of the CNIPA show that only 29 enterprises have experience in patent pledge financing. The first pledged patent was an invention patent which was registered to

CNIPA on June 25, 2010. The pledger was Suzhou Yusen New Medicine Development Co., Ltd. and the pledgee was Suzhou Sunac Guaranteed Investment Co., Ltd. To date, there have been 32 patent pledge contracts since June 25, 2010, including 112 invention patents pledged by 27 enterprises and only 15 utility model patents pledged by 2 enterprises without any design patent pledge financing. Invention patents account for 88% of pledged patents. One reason is the pharmaceutical industry's long term of industrialization and asset return, which leads to higher-risk patent pledges. The legal status of an invention patent is more stable after substantial examination, and its guarantee value is higher than that of the utility model and design patents. So, the uncertainty and risk are reduced while the rights of the pledgee can be better guaranteed. Another reason may be that there are much more invention patent applications than utility patents in pharmaceutical enterprises because the core innovations with high value are mainly protected in invention patent. Therefore, most patents pharmaceutical enterprise pledged are invention patents.

In the first three months of 2019, two patent pledge contracts have been registered, with a total financing amount of $2.6 million. Fig 2 shows the trend of the number of patent pledge financing contracts and the total amount of annual financing from 2010 to 2018. It can be seen that the number of patent pledge contracts has increased significantly since 2016, but the amount of patent pledge financing has remained. In other words, the financing amount did not increase with the increase of pledge contract. The financing amount was highest in 2013, with a total of $129.9 million, followed by a total of $30.1 million in 2017, and the amount in the remaining years was less than $28.4 million.

The patent pledge financing amount of 32 patent pledge contracts provided by the Jiangsu Intellectual Property Office shows that the minimum is $82503.6 and the maximum is $24.2 million, with the average amount being $29.9 million per contract. The pledge financing amount distribution is shown in Fig 3. It can be seen that, except for seven contracts that received financing over $4.3 million, most contracts have been small, with 56.25% receiving less than $1.4 million.

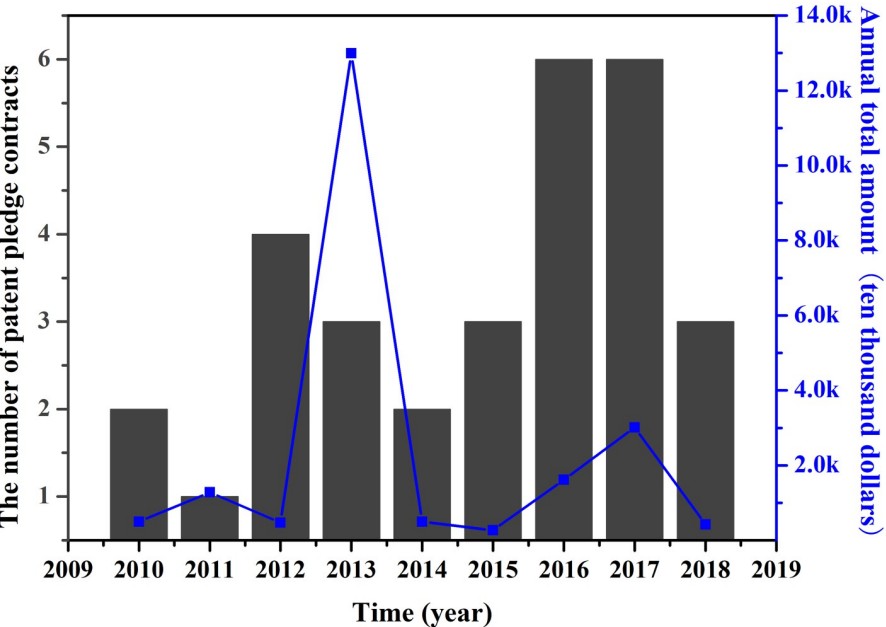

**Fig 2. The number of patent pledge contracts and the annual total amount over time.**

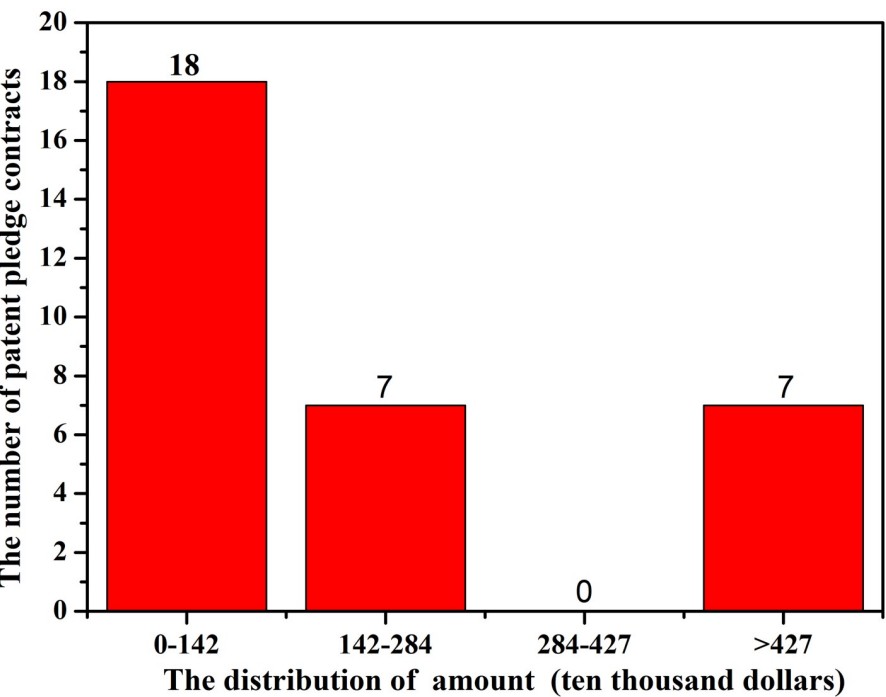

**Fig 3. The distribution of patent pledge financing amounts.**

The age of the patents is displayed in Fig 4. The patent age is calculated by the gap between the year of patent pledge and patent application. It can be seen that patent between 3–6 years are more likely to be pledged, probably because that new patents with novel technology innovations tend to be accompanied by high uncertainty and risk while old patents are out of date with no demand in the market for pledgee to turn pledged patent to tangible return.

## 4.2 Prediction model

The results of the prediction model and the calculation of the index weight of patent value were as follows:

Firstly, the indexes were compared in pairs. Saaty proposed using the numbers 1–9 and the reciprocal as the scale for judging the relative importance of the two factors (as shown in Table 1). According to this method, the judgment matrix for the patent value was constructed, as shown in Table 2.

Then, the eigenvector (formula 1) and the largest eigenvalue of the judgment matrix (formula 2) were calculated according to the following formulas:

$$W_i = \frac{\left(\prod_{j=1}^n \alpha_{ij}\right)^{\frac{1}{n}}}{\sum_{k=1}^n \left(\prod_{j=1}^n \alpha_{kj}\right)^{\frac{1}{n}}}, \tag{1}$$

$$\lambda_{\max} = \sum_{i=1}^n \frac{(A*W)_i}{n*W_i} \tag{2}$$

Thus, the eigenvector of matrix A is $W_A = (0.2876, 0.0608, 0.4762, 0.1285, 0.0469)^T$.

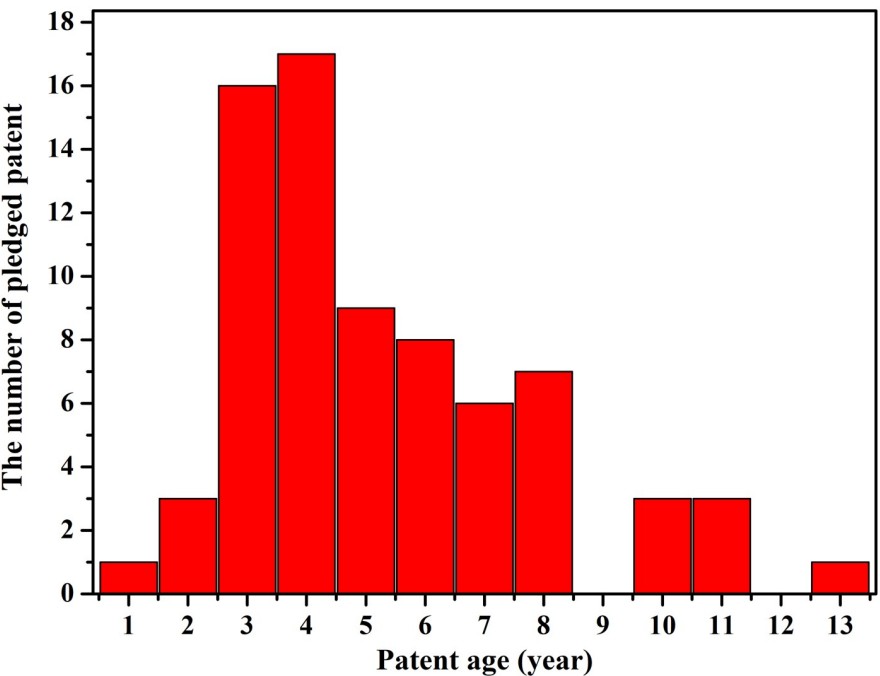

**Fig 4. The distribution of age of pledged patents.**

Largest eigenvalue of matrix A was $\lambda_{max} = 5.2789$.
The consistency index, CI, was constructed:

$$CI = \frac{\lambda_{max} - n}{n - 1} = 0.0697 \tag{3}$$

According to the formula:

$$CR = \frac{CI}{RI} \tag{4}$$

The consistency ratio (CR) was calculated. Here, the RI is a random consistency index and the order of RI with a number of alternatives is shown in Table 3 [9].

The consistency ratio of matrix A was CR = 0.0623 < 0.1. So, it passed the consistency test for matrix A and the eigenvector $W_A = (0.2876, 0.0608, 0.4762, 0.1285, 0.0469)^T$ represented the weight of the index.

**Table 1. The linguistic scale and corresponding triangular fuzzy numbers.**

| Value | Description |
|---|---|
| 1 | Indicating two equally important elements |
| 3 | Indicating one element is slightly more important than the other |
| 5 | Indicating one element is more important than the other |
| 7 | Indicating one element is strongly more important than the other |
| 9 | Indicating one element is vitally more important than the other |
| 2,4,6,8 | The median of the above adjacent judgment |
| Reciprocal | Indicating the importance of changing the order of the two factors |

**Table 2. Judgment matrix for the patent value (matrix A).**

| A | $a_1$ | $a_2$ | $a_3$ | $a_4$ | $a_5$ |
|---|---|---|---|---|---|
| $a_1$ | 1 | 5 | 1/2 | 3 | 6 |
| $a_2$ | 1/5 | 1 | 1/7 | 1/3 | 2 |
| $a_3$ | 2 | 7 | 1 | 5 | 8 |
| $a_4$ | 1/3 | 3 | 1/5 | 1 | 4 |
| $a_5$ | 1/6 | 1/2 | 1/8 | 1/4 | 1 |

According to the same method and steps, the eigenvectors and the consistency ratio of matrix O and B were calculated. The results showed that they also passed the consistency test.

In sum, the weights of indexes for the prediction of patent pledge financing amount are summarized as follows:

The weight of each index of patent pledge financing amount (O) was

$$W_O = \{0.6667, 0.3333\}.$$

The weight of each index of patent value (A) was

$$W_A = \{0.2876, 0.0608, 0.4762, 0.1285, 0.0469\}.$$

The weight of each index of the credit value of pledger (B) was

$$W_B = \{0.2109, 0.0841, 0.7049\}.$$

Combined with the index system of predicting the patent pledge financing amount we constructed above, namely, O = {A, B}, where A = {$a_1$, $a_2$, $a_3$, $a_4$, $a_5$}, B = {$b_1$, $b_2$, $b_3$}, the patent pledge financing amount could be predicted.

According to the levels and standards K = {very good, good, medium, poor, very poor} = {9,7,5,3,1}, the indexes were evaluated by the fuzzy comprehensive evaluation method. First, the first-class index, patent value (A), was evaluated. That is, each second-class index of A was evaluated separately by experts in the pharmaceutical field. Then, the corresponding scores were calculated to form the matrix of A, $G_A = [a_1, a_2, a_3, a_4, a_5]^T$. According to the weight of each index, the evaluation result of the patent value (A) was A = $W_A{}^*G_A$ = [0.2876, 0.0608, 0.4762, 0.1285, 0.0469]•[$a_1$, $a_2$, $a_3$, $a_4$, $a_5$]$^T$. Similarly, the result of the credit value of pledger (B) was B = $W_B{}^*G_B$ = [0.2109, 0.0841, 0.7049]•[$b_1$, $b_2$, $b_3$]$^T$. Based on the obtained evaluation results of A and B and their weight for the financing amount (O), the judgment matrix of O ($G_O = [A, B]^T$) was obtained and the fuzzy comprehensive prediction result of the patent pledge financing amount (O) was O = $W_O{}^*G_O$ = [0.6667, 0.3333]•[A, B]$^T$.

In this study, the patent pledge financing amount was divided into three levels: high, medium, and low amounts. The level and amount distributions are listed in Table 4.

**Table 3. The order of the random consistency index with a number of alternatives.**

| n | 1 | 2 | 3 | 4 | 5 | 6 | 7 | 8 | 9 |
|---|---|---|---|---|---|---|---|---|---|
| RI | 0 | 0 | 0.58 | 0.90 | 1.12 | 1.24 | 1.32 | 1.41 | 1.45 |

**Table 4. The level and amount distributions.**

| Level | High | Medium | Low |
|---|---|---|---|
| Amount | >6 | 3–6 | <3 |

**Table 5. Correspondence table between the prediction results and the financing amount.**

| No. | Prediction results | Financing amount | No. | Prediction results | Financing amount | No. | Prediction results | Financing amount |
|---|---|---|---|---|---|---|---|---|
| 1 | 2.0000 | 8.25 | 12 | 2.8098 | 71.12 | 23 | 5.0624 | 213.37 |
| 2 | 1.1996 | 14.22 | 13 | 2.9096 | 71.12 | 24 | 5.6348 | 260.17 |
| 3 | 2.2800 | 27.74 | 14 | 2.9960 | 71.12 | 25 | 6.7001 | 426.74 |
| 4 | 2.5139 | 28.45 | 15 | 2.5771 | 71.12 | 26 | 7.3073 | 426.74 |
| 5 | 2.5009 | 39.83 | 16 | 2.9438 | 71.12 | 27 | 6.3859 | 426.74 |
| 6 | 2.3835 | 42.67 | 17 | 2.4430 | 71.12 | 28 | 6.2883 | 426.74 |
| 7 | 2.8813 | 42.67 | 18 | 4.3670 | 142.25 | 29 | 6.1232 | 426.74 |
| 8 | 2.3024 | 42.67 | 19 | 4.1363 | 142.25 | 30 | 6.2453 | 426.74 |
| 9 | 2.4144 | 49.79 | 20 | 4.4551 | 142.25 | 31 | 8.6162 | 2418.21 |
| 10 | 2.4430 | 56.90 | 21 | 4.8032 | 152.35 | | | |
| 11 | 2.3334 | 64.01 | 22 | 4.3649 | 181.37 | | | |

The unit of financing amount is hundred thousand dollars.

## 4.3 Verification of the prediction model

The 32 patent pledge contracts provided by Jiangsu Intellectual Property Office were from 29 pharmaceutical enterprises. According to the list of enterprises that have carried out patent pledge financing, information regarding awards for science and technology and intellectual property certification was obtained from the official website of the enterprise and its credit was determined through the credit system at all levels. After excluding enterprises with missing information, 31 patent pledge contracts of 28 enterprises were used as research objects to verify the above-described model. The evaluation of experts, the relevant information on the official websites and credit system, and the pledge financing amount were used to verify the prediction model.

Taking a contract of Yancheng Jiekang Sucralose Manufacturing Co., Ltd. as an example, there were three patents in the contract and the financing amount was \$4.27 million. The experts' evaluations of the five first-class indexes of the pledged patents (A) were medium, good, good, very good, and good, forming the evaluation matrix $G_A = [5, 7, 9, 7, 7]^T$. So, the fuzzy comprehensive evaluation result of A was $A = W_A{}^*G_A = [0.2876, 0.0608, 0.4762, 0.1285, 0.0469] \cdot [5, 7, 9, 7, 7]^T = 7.3772$. The evaluation of the three first-class indexes of the credit value of the pledger (B) was good, very good, and good, forming the evaluation matrix $G_B = [7, 9, 7]^T$. So, the evaluation result of the B was $B = W_B{}^*G_B = [0.2109, 0.0841, 0.7049] \cdot [7, 9, 7]^T = 7.1675$. Correspondingly, the matrix of the patent pledge financing amount (O) was $G_O = [7.3772, 7.1675]^T$, and the prediction result was $O = W_O{}^*G_O = [0.6667, 0.3333] \cdot [7.3772, 7.1675]^T = 7.3073$.

Similarly, the prediction results of the patent pledge financing amount of the other 30 contracts were calculated, and the correspondence table between the prediction results and the financing amount is shown in Table 5. It can be observed that when the prediction result was less than 3, the financing amount of the contract was less than \$0.7 million. When the prediction result was between 4 and 6, the financing amount of the contract was between \$1.4 and \$4.3 million, and when the prediction result was greater than 6, the financing amount of the contract was more than \$4.3 million. As a consequence, the scope of the patent pledge financing amount can be predicted based on the prediction result.

## 5 Discussion and conclusions

To our knowledge, this paper is a pioneering study to investigate the prediction model of patent pledge financing amount. We described the situation of patent pledge and constructed a prediction model of patent pledge financing amount for pharmaceutical enterprises. The description analysis showed that in Jiangsu province only 5.17 percent of enterprises had experience in patent pledge and most of the pledged patents were inventions. Besides, enterprises could only receive limited financing because the financing amount is often disproportionate to the patent value. From the description, we also found that patents between 3–6 years were most suitable for pledge which means these patents were the most valuable in patent pledge financing process. As for the financing amount, results showed that it was divided into three levels: high, medium, and low amounts. Combining the prediction model and data, it can be observed that when the prediction result was less than 3, the financing amount of the contract was less than $0.7 million. When the prediction result was between 4 and 6, the financing amount of the contract was between $1.4 and $4.3 million, and when the prediction result was greater than 6, the financing amount of the contract was more than $4.3 million. As a consequence, the scope of the patent pledge financing amount can be predicted based on the prediction result.

This empirical study has several theoretical contributions. First, it enhances the knowledge of patent pledge by empirical analysis. So far consideration number of publications has investigated the definition and assessment of patent pledge, while most of them are discussed only in theory. Our work is more reliable and valid because we for the first time construct this prediction model based on systematical data from official data sources such as the Patent Pledge Registration System of the CNIPA, Jiangsu Provincial Intellectual Property Office, and Jiangsu Provincial Drug Administration. Second, our study constructs a prediction model of patent pledge financing amount for the first time which can be applied in the actual process in patent pledge. Besides, when constructing the model, the pledged patent and the pledger were both taken into consideration and a reliable patent pledge amount was used to scientifically verify the model, which makes our model more reliable, comprehensive, and innovative. Third, our work improved the knowledge of solutions to financial stress for small- and medium-sized enterprises in pharmaceutical industry.

This paper also has some practical contributions. With the prediction model of patent pledge financing amount, enterprises can estimate the financing amount by pledging certain or some patents. Also, they can obtain the desired financing amount by selecting suitable patents. In addition, the pledgee can also evaluate the enterprise and the pledged patent through the prediction model proposed in this paper. The model is also useful for patent value assessment agencies to evaluate the patent pledge financing amount. On the other hand, the Office of the Ministry of Finance together with the Office of National Intellectual Property Administration issued a notice regarding the construction of an intellectual property operation service system in 2019 and stated that the central government will support $20 million to each city for the construction, including the improvement of the pledge financing amount, the expansion of pledges and the improvement of pledge financing risk sharing. According to the analysis and the prediction model in this paper, we propose that pharmaceutical enterprises should try to cultivate high-value patents. For major technological breakthroughs, they must timely apply for patents and emphasize patent layouts. In the process of writing a patent, they should pay attention to the scope of protection and the quality of the patent as a whole, especially the claims. In addition, pharmaceutical enterprises should actively make patent transformation efforts and turn technology into economic strength. Secondly, since innovation is the foundation of development and the key to competition, enterprises should promote innovation by

focusing on basic research and technology accumulation in the process of imitative innovation. After doing so, they can gradually carry out independent innovation to improve the overall innovation ability. Finally, enterprises should attach importance to credit and abide by laws and regulations, such as repaying loans on time, to establish the good credit necessary to carry out patent pledge financing. For policymakers, they should pay attention to both the number and the quality of patent pledge to ensure that patent pledge can solve effectively financing problems. For instance, government can participate in the guarantee to reduce the uncertainty and risk for the pledgee.

Our study also has limitations. The results of the patent pledge financing amount prediction model in this paper corresponded with different financing amounts in different intervals and on the whole, the prediction results increased with the increase of the financing amount, proving that the model is scientific and reliable. However, since the evaluation of the indicators combines qualitative and quantitative analysis, there must be certain errors resulting in the prediction results in the table not strictly increasing with the increase of the financing amount. Furthermore, the prediction of patent pledge financing amount is only a range, not an accurate value. In the future, the data related to patent pledge contracts of the pharmaceutical enterprises can be collected and the relationship between the financing amount and the characteristics of patent and enterprise can be investigated.

## Supporting information

**S1 Data. The data of pledged patent.**
(XLSX)

## Author Contributions

**Conceptualization:** Yunhua Liu.

**Data curation:** Xiaojuan Zhao.

**Formal analysis:** Xiaojuan Zhao.

**Funding acquisition:** Yunhua Liu.

**Investigation:** Xiaojuan Zhao.

**Methodology:** Xiaojuan Zhao.

**Project administration:** Yunhua Liu.

**Resources:** Yunhua Liu.

**Software:** Xiaojuan Zhao.

**Supervision:** Yunhua Liu.

**Visualization:** Xiaojuan Zhao.

**Writing – original draft:** Xiaojuan Zhao.

**Writing – review & editing:** Yunhua Liu.

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
