## [Decision Letter · Decision Letter 0]

6 Apr 2020

PONE-D-20-02756

Prediction and empirical analysis of patent pledge financing amount of pharmaceutical enterprises-A case study in Jiangsu China

PLOS ONE

Dear Professor Yunhua Liu, 

Thank you for submitting your manuscript to PLOS ONE. After careful consideration, we feel that it has merit but does not fully meet PLOS ONE’s publication criteria as it currently stands. Therefore, we invite you to submit a revised version of the manuscript that addresses the points raised during the review process.

The review team appreciates your research questions and contribution. However, the consensus view that emerged was that the paper presents a review type report of regarding patent pledge financing and the theoretical discussion, structurization of the concepts, methodology, and analysis are not enough to be considered for publication in current stage. In addition, the validation of results and comparing with other studies is also required. 

We would appreciate receiving your revised manuscript by May 15 2020 11:59PM. To enhance the reproducibility of your results, we recommend that if applicable you deposit your laboratory protocols in protocols.io, where a protocol can be assigned its own identifier (DOI) such that it can be cited independently in the future. For instructions see: http://journals.plos.org/plosone/s/submission-guidelines#loc-laboratory-protocols

We look forward to receiving your revised manuscript.

Kind regards,

Wonjoon Kim, Ph.D

Academic Editor

PLOS ONE

Journal Requirements:

Reviewers' comments:

Reviewer's Responses to Questions

**Comments to the Author**

1. Is the manuscript technically sound, and do the data support the conclusions?

Reviewer #1: Yes

Reviewer #2: Yes

Reviewer #3: Yes

2. Has the statistical analysis been performed appropriately and rigorously? 

Reviewer #1: Yes

Reviewer #2: Yes

Reviewer #3: Yes

3. Have the authors made all data underlying the findings in their manuscript fully available?

Reviewer #1: Yes

Reviewer #2: Yes

Reviewer #3: No

4. Is the manuscript presented in an intelligible fashion and written in standard English?

Reviewer #1: Yes

Reviewer #2: No

Reviewer #3: Yes

5. Review Comments to the Author

Reviewer #1: in my view of point the analysis of data and the statistical for the financing amount for a patent pledge works so well to convince me and have been conducted rigorously, with appropriate controls, replication, and sample sizes , and really successful for more researching in financial field , nice work !

Reviewer #2: The study “Prediction and empirical analysis of patent pledge financing amount of pharmaceutical enterprises-A case study in Jiangsu China” is interesting. However, this paper is poorly organized with typos which makes it is very hard to read. Thus, I would recommend the authors rewritten the paper and add more explanations and also describe the innovation of this paper in the introduction section. Moreover, attention should be given to the following highlighted points before resubmitting.

Major

More recent references should be added to broaden the view of readers and enhance the new contribution of this paper for comparison.

Page 13 line 245. Table 3. The corresponding value of the judgment matrix A (row a5 and column a2 ) is not correct.

Page 14 line 248. If the same matrix A used for computation which is reported in this study, then the whole computation may be revised because the results may be affected by changing the value in A reported in 2. For instance, “ ① All elements of each row of the matrix A were multiplied and the nth root was taken.”

Page 15 line 263, How the values of RI computed please provide some information. If used already procedure than provide the reference.

Minor

Page 4 line No. 81 CNIPA stands for? First defined the word then use that abbreviation throughout the paper.

Page 6 line No. 111 please provide references to the following statements.

1. Under the promotion of national and local policies, the number of patent pledge contracts has increased. However, the amount of patent pledge financing is still low.

2. The evaluated value of pharmaceutical patents for pledge financing is far lower than their actual value because there is no unified evaluation standard and the pharmaceutical industry is characterized by rapid technology substitution and high risk.

Page 14 line 254. ② The largest eigenvalue was calculated according to the formula: λ_max=∑_(i=1)^n▒〖(AW)〗_i/(nW_i )

What is AW in the above formula? How to compute AW?

Page 14 line 254, Remove the repetition of the formulas λ_max=∑_(i=1)^n▒〖(AW)〗_i/(nW_i ) , Line 257, CI=(λ_max-n)/(n-1)=0.0697. and line CR=CI/RI

There is some punctuation problem.

Reviewer #3: 1. Author(s) here tried to present a review type report of regarding patent pledge financing, pharmaceutical enterprises, etc. It is a research article with some new information and confusions. However, I have few yet suggestively serious reservations:

2. First of all, title of the paper is overloaded with the heavy keywords, it is hard to grasp for non-technical (out of field) reader.

3. Highlights are missing. I am not sure if that was optional. Usually it is given in such sort of studies.

4. Abstract does not portray any novelty, impressive results, and workable policy implications.

5. Literature review is another concern, improvement needed in this section as well. Extensive round of literature is suggested.

6. Page 5, second paragraph, mention currency unit and equivalence to USD in footnotes.

7. Section 2, 3, try to add standard sub-sections instead of numbering them.

8. Another concern is related to methodology section, I have following issues in this:

9. What sort of data is used? Mostly sources are mentioned.

10. I am unsure if sections 3 and 4 belong to methods.

11. In any case, sections 3 and 4 are too lengthy. Please reduce information in those as much as possible.

12. Why there is no proper section containing results and discussion?

13. How about validation of results and comparing them with other studies (not necessary to quote the same but in similar areas)?

14. Too many tables and few figures, two probably. It is imbalance of presentation. Please balance both of them. Try to convert few tables into figures or at least add more figures.

15. Page 19, lines 349-343, repetition of information, please remove or replace.

16. It is useless to say revise conclusion as well because after all such revisions, mentioned above, conclusion would be revised and improved.

17. How conclusions and prospects are different?

6. PLOS authors have the option to publish the peer review history of their article (what does this mean?). If published, this will include your full peer review and any attached files.

Reviewer #1: No

Reviewer #2: No

Reviewer #3: Yes: Ghaffar Ali

---

## [Author Response · Author response to Decision Letter 0]

15 Apr 2020

Response to Reviewer 1: 

Reviewer #1: in my view of point the analysis of data and the statistical for the financing amount for a patent pledge works so well to convince me and have been conducted rigorously, with appropriate controls, replication, and sample sizes, and really successful for more researching in financial field , nice work !

Answer: We are very grateful for your time and this comment.

Response to Reviewer 2: 

Reviewer #2: The study “Prediction and empirical analysis of patent pledge financing amount of pharmaceutical enterprises-A case study in Jiangsu China” is interesting. However, this paper is poorly organized with typos which makes it is very hard to read. Thus, I would recommend the authors rewritten the paper and add more explanations and also describe the innovation of this paper in the introduction section. Moreover, attention should be given to the following highlighted points before resubmitting.

Major

More recent references should be added to broaden the view of readers and enhance the new contribution of this paper for comparison.

Answer: Thanks for this valuable comment. We have added some recent references for the comprehensibility and contributions of this manuscript in the section of the literature review in the revised manuscript.

Page 13 line 245. Table 3. The corresponding value of the judgment matrix A (row a5 and column a2 ) is not correct.

Answer: Thanks for pointing out this. We have revised and corrected this clerical error in the revised manuscript.

Page 14 line 248. If the same matrix A used for computation which is reported in this study, then the whole computation may be revised because the results may be affected by changing the value in A reported in 2. For instance, “ ① All elements of each row of the matrix A were multiplied and the nth root was taken.”

Answer: Thank you for this suggestion. We have revised the whole computation. It showed that the mistake in table 3 was just clerical error and the results were based on the correct values.

Page 15 line 263, How the values of RI computed please provide some information. If used already procedure than provide the reference.

Answer: Thank you for pointing it out. The values of RI are quoted from the following literature:

Saaty T. The analytic hierarchy process McGraw-Hill. New York, 1980, 324.

Minor

Page 4 line No. 81 CNIPA stands for? First defined the word then use that abbreviation throughout the paper.

Answer: We are very grateful for your advice. We have corrected it and the abbreviation was defined when it first appeared in the revised manuscript.

Page 6 line No. 111 please provide references to the following statements.

1. Under the promotion of national and local policies, the number of patent pledge contracts has increased. However, the amount of patent pledge financing is still low.

2. The evaluated value of pharmaceutical patents for pledge financing is far lower than their actual value because there is no unified evaluation standard and the pharmaceutical industry is characterized by rapid technology substitution and high risk.

Answer: Thanks for this suggestion. References have been added to the statements in the revised manuscript.

Page 14 line 254. ② The largest eigenvalue was calculated according to the formula: λ_max=∑_(i=1)^n▒〖(A*W)〗_i/(nW_i )

What is AW in the above formula? How to compute AW?

Answer: Thanks for this valuable comment. A is the matrix and W is the eigenvector of matrix A. AW means matrix multiplied by eigenvector. We have expressed it more clearly in the revised manuscript.

Page 14 line 254, Remove the repetition of the formulas λ_max=∑_(i=1)^n▒〖(AW)〗_i/(nW_i ) , Line 257, CI=(λ_max-n)/(n-1)=0.0697. and line CR=CI/RI

There is some punctuation problem.

Answer: Thanks for pointing them out. We have made corrections in the revised manuscript.

Response to Reviewer 3: 

Reviewer #3: 1. Author(s) here tried to present a review type report of regarding patent pledge financing, pharmaceutical enterprises, etc. It is a research article with some new information and confusions. However, I have few yet suggestively serious reservations:

Answer: We are very grateful for your time and comments. We have revised the manuscript overall according to your suggestions, with details as follows.

2. First of all, title of the paper is overloaded with the heavy keywords, it is hard to grasp for non-technical (out of field) reader.

Answer: Thank you for raising this question. The title has been simplified for all kind of readers as follows:

Empirical Prediction of patent pledge financing of pharmaceutical enterprises -A case study in Jiangsu China

3. Highlights are missing. I am not sure if that was optional. Usually it is given in such sort of studies.

Answer: Thank you for this suggestion. Highlights have been added in the revised manuscript as follows:

We construct the prediction model of patent pledge financing amount for the first time which can be applied in the actual process in patent pledge.

This work is more reliable with real data of pledged patents and the corresponding financing amount from official data sources.

This study provides guides for pledgees and policymakers to improve the efficiency and quality of patent pledge.

4. Abstract does not portray any novelty, impressive results, and workable policy implications.

Answer: Thank you for spotting this for us. We have rewritten abstract by adding the novelty, impressive results, and workable policy implications in the revised manuscript.

5. Literature review is another concern, improvement needed in this section as well. Extensive round of literature is suggested.

Answer: Thanks for this valuable comment. We have revised the literature review and we also cited more relevant literature to summarize the trend in patent pledge research in the revised manuscript.

6. Page 5, second paragraph, mention currency unit and equivalence to USD in footnotes.

Answer: Thanks for the reviewer’s valuable suggestion. Instead of mentioning the currency unit and equivalence to USD in footnotes, we used dollar ($) directly in the revised manuscript. 

7. Section 2, 3, try to add standard sub-sections instead of numbering them.

Answer: Thank you so much for helping us improve the manuscript. We agree and have reordered and numbered the sections in the revised manuscript.

8. Another concern is related to methodology section, I have following issues in this:

Answer: Thanks for the following suggestions. We have revised this section accordingly in the revised manuscript.

9. What sort of data is used? Mostly sources are mentioned.

Answer: Thanks for pointing it out. Data sources have been presented in section 3.1 in the revised manuscript.

10. I am unsure if sections 3 and 4 belong to methods.

Answer: Thanks for this valuable comment. We have reordered and numbered the sections which could make the manuscript more readable and precise.

11. In any case, sections 3 and 4 are too lengthy. Please reduce information in those as much as possible.

Answer: Thanks for the valuable suggestion. We have reduced information and made it more concise and precise in these sections in the revised manuscript.

12. Why there is no proper section containing results and discussion?

Answer: Thank you for pointing it out. We have revised the manuscript and reorganized it with the right structure in the revised manuscript.

13. How about validation of results and comparing them with other studies (not necessary to quote the same but in similar areas)?

Answer: We are very grateful for this comment. We have added the analysis in the section of discussion and conclusions in the revised manuscript.

14. Too many tables and few figures, two probably. It is imbalance of presentation. Please balance both of them. Try to convert few tables into figures or at least add more figures.

Answer: We are very grateful for your advice. In the revised manuscript we have balanced the tables and figures. In detail, we convert the first table into figure, deleted two tables, and add a figure to describe the situation of patent pledge in Jiangsu Province.

15. Page 19, lines 349-343, repetition of information, please remove or replace.

Answer: Thanks for this valuable comment. We have extracted this section in the revised manuscript.

16. It is useless to say revise conclusion as well because after all such revisions, mentioned above, conclusion would be revised and improved.

Answer: Thank you for this suggestion. We have revised and improved the section of discussion and conclusions. Summary, theoretical contributions, practical contributions, and limitations are included in this section in the revised manuscript

17. How conclusions and prospects are different?

 Answer: Thanks for pointing it out. Actually, it was wordy at the end of the manuscript. We have revised these sections and extracted them in the discussion and conclusions in the revised manuscript.

---

## [Decision Letter · Decision Letter 1]

11 May 2020

Empirical prediction of patent pledge financing of pharmaceutical enterprises -A case study in Jiangsu China

PONE-D-20-02756R1

Dear Dr. Yunhua Liu,

We are pleased to inform you that your manuscript has been judged scientifically suitable for publication and will be formally accepted for publication once it complies with all outstanding technical requirements.

With kind regards,

Wonjoon Kim, Ph.D

Academic Editor

PLOS ONE

Reviewers' comments:

Reviewer's Responses to Questions

**Comments to the Author**

1. If the authors have adequately addressed your comments raised in a previous round of review and you feel that this manuscript is now acceptable for publication, you may indicate that here to bypass the “Comments to the Author” section, enter your conflict of interest statement in the “Confidential to Editor” section, and submit your "Accept" recommendation.

Reviewer #1: All comments have been addressed

Reviewer #2: All comments have been addressed

Reviewer #3: All comments have been addressed

2. Is the manuscript technically sound, and do the data support the conclusions?

Reviewer #1: Yes

Reviewer #2: Yes

Reviewer #3: Yes

3. Has the statistical analysis been performed appropriately and rigorously? 

Reviewer #1: Yes

Reviewer #2: Yes

Reviewer #3: Yes

4. Have the authors made all data underlying the findings in their manuscript fully available?

Reviewer #1: Yes

Reviewer #2: Yes

Reviewer #3: Yes

5. Is the manuscript presented in an intelligible fashion and written in standard English?

Reviewer #1: Yes

Reviewer #2: Yes

Reviewer #3: Yes

6. Review Comments to the Author

Reviewer #1: (No Response)

Reviewer #2: All of my comments from the previous review round have been incorporated so i am accepting the paper in the current form.

Reviewer #3: Changes are satisfactory. Authors have addressed all the comments and suggestions asked previously. Now it is ready for publication.

7. PLOS authors have the option to publish the peer review history of their article (what does this mean?). If published, this will include your full peer review and any attached files.

Reviewer #1: Yes: Haitham Medhat Aboulilah

Reviewer #2: No

Reviewer #3: Yes: Dr. Ghaffar Ali, Shenzhen University, Shenzhen, China

---

## [Editor Report · Acceptance letter]

20 May 2020

PONE-D-20-02756R1 

Empirical prediction of patent pledge financing of pharmaceutical enterprises -A case study in Jiangsu China 

Dear Dr. Liu:

I am pleased to inform you that your manuscript has been deemed suitable for publication in PLOS ONE. Congratulations! Your manuscript is now with our production department. 

With kind regards,

on behalf of

Dr. Wonjoon Kim 

Academic Editor

PLOS ONE